# The Hypolipidemic and Antioxidant Activity of Wheat Germ and Wheat Germ Protein in High-Fat Diet-Induced Rats

**DOI:** 10.3390/molecules27072260

**Published:** 2022-03-31

**Authors:** Cong Liu, Yi Sun, Lei Yang, Yuxian Chen, Rigala Ji, Hao Wang, Jinghong Shi, Jilite Wang

**Affiliations:** 1Department of Agriculture, Hetao College, Bayannur 015000, China; liucong520wang@163.com (C.L.); sunyi19940123@163.com (Y.S.); jrgl8392@126.com (R.J.); 2Bayannaoer Academy of Agricultural and Animal Sciences, Bayannur 015000, China; yanglei0301@126.com; 3Inner Mongolia Hengfeng Group Yinliang Flour Industry Co., Ltd., Bayannur 015000, China; chenyuxian800@sohu.com; 4State Key Laboratory of Food Nutrition and Safety, Tianjin University of Science and Technology (TUST), Tianjin 300457, China; wanghao@tust.edu.cn

**Keywords:** wheat germ, wheat germ protein, hypolipidemic, antioxidant

## Abstract

Background: So far, no articles have discussed the hypolipidemic effect of wheat germ protein in in vivo experiments. Objective: In this study, we investigated the effects of wheat germ protein (WGP, 300 mg/kg/day) and wheat germ (WG, 300 mg/kg/day) on cholesterol metabolism, antioxidant activities, and serum and hepatic lipids in rats fed a high-fat diet through gavage. Methodology: We used 4-week-old male Wistar 20 rats in our animal experiment. Biochemical indicators of fecal, serum and liver were tested by kits or chemical methods. We also conducted the cholesterol micellar solubility experiment in vitro. Results: After 28 days of treatment, our results showed that WGP significantly reduced the serum levels of total cholesterol (*p* < 0.05) and nonhigh-density lipoprotein cholesterol (*p* < 0.05), improved the enzymatic activities of cholesterol 7-α hydroxylase (*p* < 0.01) and low-density lipoprotein receptor (*p* < 0.01) and increased bile acid excretion in feces (*p* < 0.05). Conclusion: WG did not significantly increase bile acid excretion in feces or decrease serum levels of total cholesterol. Moreover, WGP and WG both presented significant antioxidant activity in vivo (*p* < 0.05) and caused a significant reduction in cholesterol micellar solubility in vitro (*p* < 0.001). Therefore, WGP may effectively prevent hyperlipidemia and its complications as WGP treatment enhanced antioxidant activity, decreased the concentration of serum lipids and improved the activity of enzymes involved in cholesterol metabolism.

## 1. Introduction

Wheat (*Triticum aestivum* L.) is one of the three most common grains and is widely planted worldwide. The caryopsis of wheat is a staple food for human beings. After being milled into flour, the caryopsis can be used to make bread, steamed bread, biscuits, noodles and other foods [1]. Therefore, wheat is produced and processed in large quantities. Wheat germ (WG) is a byproduct that can be separated from wheat during milling. WG contains approximately 10–15% lipids, 26–35% protein, 17% sugar, 1.5–4.5% fiber and about 4% minerals [2]. Naturally, the germ can be utilized in many areas such as for food, pharmaceutical and other biological purposes [3]. In recent articles, many researchers have explored the hypolipidemic, antiaging, antibacterial and antioxidant effects of wheat germ [4,5,6,7]. These results indicate that WG as a biological substance should be studied in depth to examine the various biologically active ingredients contained in it.

The antioxidant activity and total phenolic contents of defatted wheat germ have been reported [8]. Due to its N-acetylglucosamine affinity, several studies have demonstrated that wheat germ agglutinin could be a useful compound for various biomedical and therapeutic applications, such as chemotherapy, targeted drug delivery, antibiotic-resistant bacteria monitoring and elimination [9]. In several papers, wheat germ polysaccharides have been shown to regulate the immune response of mice associated with intestinal microbiota and have antioxidant activity [10,11]. Wheat germ oil (WGO) includes fatty acids, lipids, unsaponifiable lipids, tocopherols, n-alkanols, hydrocarbons, pigments, sterols, 4-methyl sterols, triterpenes and other nutrients, among which the most important contents are fatty acids and vitamin E [12]. In recent years, some studies have stated that WGO can prevent the production of oxidized lipids, inhibit free radicals, slow body aging, protect against inflammation, and lower cholesterol and be used to treat circulatory and cardiac disorders and weaknesses [13]. WGO is one of the few antioxidants that are truly effective in promoting antiaging in humans. In previous studies, the main proteins of wheat germ protein (WGP) were found to be albumin, globulin, prolamin, and gluten [14]. Therefore, WGP is rich in essential amino acids, especially methionine, lysine and threonine. This means that WGP is an attractive and promising plant protein resource. To date, research on WGP and WGP hydrolysates has focused mainly on immunity improvement, gut microbiota remodeling, and antioxidant and free-radical-scavenging activities [14,15]. In our paper, we first explored the hypolipidemic effect of WGP through animal experiments. 

A high-calorie, high-cholesterol, high-saturated fatty acid diet, obesity, aging and lipid metabolic disturbance are the main causes of hyperlipidemia [16]. Hyperlipidemia is characterized by an increase in total cholesterol (TC), triglycerides (TG) and low-density lipoprotein cholesterol (LDL-c) and a decrease in high-density lipoprotein cholesterol (HDL-c) [17]. Hyperlipidemia is a major risk factor for the development of arteriosclerosis, cardiovascular diseases and their complications. Annually, more than four million people die because of hyperlipidemia. Considering the side effects of pharmacological treatment, searching for bioactive compounds that are contained in foods and that lower blood lipids has attracted much attention from researchers. In this research, by detecting the concentration of TC, TG and HDL, the activity level of 3-hydroxy-3-methyl glutaryl-coenzyme A (HMG-CoA) reductase, cholesterol 7-α hydroxylase (CYP7A1) and low-density lipoprotein receptor (LDL-R) and the concentration of antioxidant ability indicators, we further characterized the hypolipidemic effect of WGP in high-fat diet-induced rats.

## 2. Results

### 2.1. Effect of WG and WGP Treatment on Growth Biomakers

As shown in Table 1, the initial body weight, body weight gain and food intake were not significantly different among the normal control (NC), high-fat (HF), wheat germ (WG) and wheat germ protein (WGP) groups. However, the final body weight of the WGP group was similar with HF and WG groups and higher than that of the NC group (*p* < 0.05). The liver relative weight of the NC group was significantly lower than that of the HF, WG and WGP groups (*p* < 0.01). Therefore, by comparing with NC group, a high-fat diet alters the growth of rats and the WG and WGP treatment do not alter the growth of high-fat diet-induced rats. 

### 2.2. Effect of WG and WGP Treatment on Serum Lipid Profiles

As shown in Figure 1, treatment with WGP caused a significant reduction in serum cholesterol compared with the HF group (*p* < 0.05). No significant difference was observed in TG between the NC group, HF group, WG group and WGP group. No significant difference was observed in HDL-c between the NC group, HF group, WG group and WGP group. WGP intake significantly lowered non-HDL-c levels in the high-fat-diet rats compared with the HF group (*p* < 0.05). Under the treatment of WGP, the levels of TC and non-HDL-c returned to a normal level.

### 2.3. Effect of WG and WGP Treatment on Liver Lipid Profiles

As shown in Figure 2, the contents of TC, TG and total lipids in the livers of high-fat diet-induced rats were significantly higher than rats fed with normal diet (*p* < 0.001). The administration of WG and WGP did not significantly affect the content of TC, TG or total lipids in rat liver compared with the HF group. However, after treatment with WG and WGP, the level of TC was found to be slightly higher than that in the HF group.

### 2.4. Effect of WG and WGP Treatment on Enzymatic Activity Related to Cholesterol Metabolism

As shown in Figure 3, WG and WGP intake both had a significant effect on the concentration of CYP7A1 in the liver compared with the HF and NC groups (*p* < 0.01). However, the concentration of HMG-CoA reductase in the WG and WGP groups was not significantly higher than that in the HF and NC groups. Moreover, the WG and WGP treatments also significantly increased the concentration of LDL-R compared with the HF and NC groups (*p* < 0.01). WG showed better results than WGP (*p* < 0.01). In comparison with the NC group, the high-fat diet cannot affect the enzyme activity of CYP7A1, HMG-CoA and LDL-R.

### 2.5. Effect of WG and WGP Treatment on Fecal Matter

As shown in Figure 4A, fecal excretion was significantly higher in the WGP groups (*p* < 0.05) than in the HF group. Moreover, fecal excretion was significantly higher in the NC group than the HF, WG and WGP groups (*p* < 0.01). As shown in Figure 4B, no significant changes were observed in fecal cholesterol, although the fecal cholesterol content was slightly higher in the WG and WGP groups than in the HF group. As shown in Figure 4C, the fecal bile acid content in the WGP group was significantly higher than that in the NC (*p* < 0.01) and HF groups (*p* < 0.05).

### 2.6. Effect of WG and WGP on Cholesterol Micellar Solubility In Vitro

As shown in Figure 5, the administration of both WG and WGP significantly decreased the micellar solubility of cholesterol in vitro, compared with the administration of casein (*p* < 0.001). Treatment with WGP may have a better effect in terms of inhibiting cholesterol absorption.

### 2.7. Effects of WG and WGP on the Antioxidant Index of Hyperlipidemic Rats

Figure 6 shows that both WG and WGP significantly improved the activities of serum SOD (*p* < 0.01), T-AOC, and CAT and reduced the level of serum MDA (*p* < 0.01) in high-fat diet-induced rats after 4 weeks compared with the HF group. The treatment of WG and WGP significantly improved the activity of serum SOD (Figure 6A) and T-AOC (Figure 6B) compared with the NC group. Comparing the activity of serum CAT with the NC group, the WGP group was significantly higher than that of NC group (Figure 6D, *p* < 0.01), but the WG group was similar to the NC group. The levels of serum MDA in the WG and WGP groups were significantly higher than that of NC group (Figure 6E, *p* < 0.01). However, treatment with either WG or WGP did not significantly change the activity of serum GSH-Px (Figure 6C) compared with the NC and HF groups. For the serum antioxidant index, the administration of WGP showed a better effect.

Figure 7 shows the activities of liver SOD, T-AOC, GSH-Px, and CAT and the level of liver MDA in high-fat diet-induced rats. The level of liver T-AOC in the HF groups was significantly lower than that of the NC group (Figure 7B, *p* < 0.05), and the administration of WGP induced a significant effect on T-AOC compared with the HF group (Figure 7B, *p* < 0.05). For the level of GSH-Px, the HF, WG and WGP groups were significantly higher than NC group (Figure 7C, *p* < 0.05), but no significant difference was observed in the HF, WG and WGP groups. The activity of SOD was significantly increased under the treatment of both WG and WGP (Figure 7A, *p* < 0.01) compared with the NC group and HF group. Only WG significantly increased the activity of CAT (Figure 7D, *p* < 0.05) compared with the NC group and HF group. The level of CAT in WGP group was significantly higher than in the NC group (Figure 7D, *p* < 0.01) and similar with the HF and WG groups. The level of MDA in the liver decreased significantly with the WGP or WG treatment (Figure 7E, *p* < 0.05) compared with HF. However, the level of MDA in WGP group was significantly higher than in the NC group (Figure 7E, *p* < 0.01). For the liver antioxidant index, the administration of WG showed a better effect.

## 3. Discussion

It is well known that the increased level of serum cholesterol or lipid profiles can promote atherosclerosis and other cardiovascular diseases [18]. An excess of cholesterol is used in most research to cause rapid hypercholesterolemia and atherosclerosis [19]. Traditional strategies for regulating hypercholesterolemia either decrease in bad cholesterol, i.e., LDL-cholesterol, or increase in good cholesterol, i.e., HDL-cholesterol [20,21].

In our research, we explored the hypolipidemic effect of wheat germ and wheat germ protein on high-fat-diet rats in vivo and in vitro. The administration of WG and WGP can effectively prevent metabolic complications in high-fat-diet rats with improvements in fecal and fecal bile acid excretion, a reduction in serum and liver lipid profiles, an increase in the protein levels involved in cholesterol metabolism, and modulate the antioxidant protein levels of serum and liver. In particular, we evaluated the inhibitory effect of WG and WGP on cholesterol micellar solubility. 

In previous studies, a significant negative correlation was found between fecal steroid excretion and serum total cholesterol in rats [22]. In our study, the treatment of WG and WGP can significantly increase the excretion of fecal matter compared to the HF group (Figure 4A, *p* < 0.05). Furthermore, WGP significantly promoted the excretion of fecal bile acids (Figure 4C, *p* < 0.05). The classic bile acid’s biosynthetic pathway is generated by CYP7A1 by converting cholesterol to bile acid in the liver [23]. Thus, the stimulation of the excretion of fecal and fecal bile acids are two cholesterol-lowering actions of WGP. 

Many studies have stated that plasma cholesterol levels can be affected by dietary lipids [24]. The uptake cholesterol is first emulsified in the jejunal lumen under the function of bile salt and lecithin and then encapsulated into micelles. Micelles are water-soluble polymolecule that can facilitate cholesterol uptake into mucosal cells [25]. So, artificial micelles have been used as a model system for evaluating natural products aimed at lowering plasma cholesterol levels in many researches [26]. In our research, we discovered that WG and WGP can significantly suppress the micellar solubility of cholesterol compared with casein and WGP has a better effect (Figure 5, *p* < 0.01). This result indicates that WG and WGP can effectively inhibit the absorption of cholesterol in intestinal. For better illustrating the cholesterol-lowing effect of WG and WGP in the intestines, we will explore the bile acid-binding ability and cholesterol absorption in Caco-2 cells in a future study.

Moreover, we discovered that the administration of WG and WGP results in significant reductions in serum total cholesterol (TC) and non-HDL-cholesterol concentrations in hypercholesterolemic rats fed a high-cholesterol diet (Figure 1, *p* < 0.05). In a previous study, the treatment of wheat germ oil to hypercholesterolemic rabbits caused a significant reduction in serum cholesterol and non-HDL-cholesterol, in agreement with our results [27]. Our data were also consistent with the hypocholesterolemia effect of milk whey [28], oat protein [29] and egg white protein [30]. Obviously, WGP has a better effect than WG in terms of reducing serum TC and non-HDL-c. Thus, selecting intact proteins or bioactive peptides from wheat germ protein will play pivotal role in future studies.

However, the administration of WG and WGP did not significantly change the concentrations of hepatic cholesterol, hepatic triglycerides and hepatic lipids (Figure 2). Thus, we explored the enzyme concentration related to cholesterol mechanism in the liver.

CYP7A1 as a rate-limiting enzyme regulates the cholesterol conversion to bile acid in the liver [31]. Bile acids transform from cholesterol via intestinal and biliary lumen excreted from the body. So, promoting the activity of CYP7A1 accelerates the generation rating of bile acids [18]. In present study, our data stated that the treatment of WG and WGP significantly increased the concentration of CYP7A1 in hypercholesterolemic rats fed a high-fat diet (Figure 3, *p* < 0.01). This result accords with our fecal experiments. These findings suggested that WG and WGP may enhance the conversion of cholesterol into bile acids and fecal excretion to promote the circulation of cholesterol. Cholesterol biosynthesis is a complex process involving more than 30 enzymes, and HMG-CoA reductase as a rate-limiting enzyme plays a pivotal role in the cholesterol biosynthetic pathway. So, inhibiting the activity and concentration of HMG-CoA reductase is one possible method to reduce cholesterol biosynthesis [32]. In our study, the treatment of WG and WGP did not significantly increase the concentration of HMG-CoA reductase compared with the high-fat group (Figure 3). Although WG and WGP showed the ability to enhance the activity of HMG-CoA reductase, the content of TC in the liver is stable among HF, WG and WGP groups (Figure 2). This finding indicates that WG and WGP may not decline the content of cholesterol on this way. LDL-R is localized at the basolateral membrane and is responsible for acquiring cholesterol in the form of LDLs from the blood. LDL-R is a main player in regulating plasma cholesterol level and cholesterol homeostasis by limiting the hepatic uptake of circulating cholesterol [18,33]. Our data showed that the administration of WG and WGP significantly elevated the content of LDL-R (Figure 3, *p* < 0.01). This result means that hepatocytes would uptake more LDL-c in serum than the high-fat diet group. Additionally, this result also accords with previous data that show that total cholesterol in serum decreased significantly (Figure 2, *p* < 0.05). These findings suggested that WG and WGP could lower the serum lipids by increasing the content of LDL-R. 

Many studies have clearly reported that high-fat diets can produce a great amount of free radicals and when their production exceeds the self-scavenging ability [34]. Due to the accumulation of reactive oxygen species (ROS) free radicals, the body will be in an oxidative stress state. Previous studies stated that ROS could form ox-LDL-C by oxidizing LDL-C and this complex is closely related to the formation of hyperlipidemia [35]. Malondialdehyde (MDA) is one of the final products of polyunsaturated fatty acid peroxidation in cells. MDA can cause the cross-linking of proteins and nucleic acids and is cytotoxic. Thus, the content of MDA is an important indicator of reflecting the body’s antioxidation potential and can also indirectly reflect the degree of tissue peroxidation damage [11]. In our study, we found that the treatment of WG and WGP both significantly decreased the content of MDA in serum (Figure 6E) and the treatment of WGP also significantly decreased the content of MDA in the liver (Figure 7E). In order to prevent excessive oxidation in the body, the enzyme antioxidant system plays a key role in maintaining the balance of oxidative metabolism. The enzyme antioxidant system mainly includes superoxide dismutase (SOD), glutathione peroxidase (GSH-Px), catalase (CAT); etc. SOD is an important antioxidant enzyme, which can effectively convert free radicals (superoxide O2- and hydroxyl OH-) to H_2_O_2_. GSH-Px is an endogenous antioxidant, which can decompose peroxides by converting H_2_O_2_ into H_2_O and CO_2_ [36]. Thus, GSH-Px inhibits the synthesis of MDA in the body. CAT contains four ferriprotoporphyrin groups per molecule and plays a crucial role in adaptive response to H_2_O_2_ and in the adaptive response to proton donors (ROOH). CAT somehow assists GSH-Px by converting H2O2 and ROOH into H_2_O [37]. The sum of antioxidant activities of less specific antioxidants and nonspecific antioxidants is usually called the total antioxidant capacity (T-AOC). T-AOC includes various hydrophilic and hydrophobic low-molecular-weight substances such as glutathione, ascorbic acid (vitamin C), uric acid, tocopherols (vitamin E), carotenoids, coenzyme Q, bilirubin, some amino acids (such as cysteine, methionine, or tyrosine) [38], etc. In the present study, we found that the treatment of WG and WGP can significantly increase the concentration of SOD in the liver (Figure 7A, *p* < 0.01). WG also has a significant effect on increasing the content of CAT in the liver (Figure 7D, *p* < 0.05), while WGP significantly decreased the concentration of MDA the in liver (Figure 7E, *p* < 0.05). For serum, the SOD, T-AOC and CAT levels of WG and WGP groups had a significantly improvement compared with the HF group (*p* < 0.05). Furthermore, the MDA level in serum was significantly reduced under the treatment of WG and WGP (*p* < 0.05). These results indicated that WG and WGP are potential biosubstances that enhance the body’s antioxidant capacity. Moreover, WGP showed a better antioxidant capacity in serum.

## 4. Materials and Methods

### 4.1. Preparation of Wheat Germ Powder and Wheat Germ Protein Powder

Wheat (*Triticum aestivum* L.) germ was purchased from the Inner Mongolia Hengfeng Food Industry (Group) Co., Ltd. (Inner Mongolia Autonomous Region, China) The wheat germ was cleaned and then turned into powder by passing it through a 60-mesh sieve. WGP was extracted from the wheat germ by alkali solution and acid isolation methods and then milled into powder. Wheat germ was dissolved in distilled water in a 2 L beaker to create a 10% solution, and then stirred for 2 h at 350 r/m at 50 ℃ with a constant temperature magnetic stirrer (SCILOGEX, Hartford, CT, USA). The pH value was adjusted to 9.5 by 0.1 mol/L NaOH, and then, the solution was centrifuged at 3000× *g* for 20 min to obtain the supernatant. The pH of the supernatant was adjusted to pH 4.5 with 0.1 mol/L HCl, incubated for 2 h, and then centrifuged at 3000× *g* for 20 min to obtain the precipitate. The precipitate was washed with distilled water, and then the pH value was adjusted to pH 7.0. Finally, the residue was dried with a vacuum freeze dryer (Beijing Bo Yikang Experimental Instrument Co., Ltd., Bejing, China). The purity of the WGP was 80.10%.

### 4.2. Animal Experiment

Four-week-old, male Wistar rats weighing 70~80 g were obtained from Sibefu (Beijing) Biotechnology Co. Ltd. The rats were housed in individual cages under a 12 h light–12 h dark cycle under temperature-controlled conditions at 22 ± 2 °C, with free access to food and water. After one week of being fed a commercial unpurified diet, the rats were divided into four groups. Animal procedures were approved by the Ethics Committee for Animal Research of the Hetao College (SCXK 2019-0010, 28 October 2020).

Finally, twenty rats were divided into four experimental groups (*n* = 5 in each group) based on body weight: normal control (NC) group, high-fat diet (HF) group, WG group and WGP group. The amino acid compositions of the WG and WGP are shown in Table 2. The amino acid compositions of WG and WGP were detected by L-8900 Amino Acid Analyzer (Hitachi, Japan). HF, WG and WGP groups had access to a high-cholesterol diet containing casein during the experimental period. The composition of the high-cholesterol diet was as follows: casein, 20%; lard, 5%; corn oil, 1%; AIN93 mineral mix, 3.5%; AIN93 vitamin mix, 1%; choline chloride, 0.2%; cellulose, 5%; cholesterol, 1%; sodium cholate, 0.5%; sucrose, 20.9%; and cornstarch, 41.9%. NC group had free access to a normal diet containing casein(no cholesterol and sodium cholate)during the experimental period. The WG powder and WGP powder (300 mg/kg B.W/day) were dissolved in 0.5% carboxymethyl-cellulose sodium salt (Sigma Aldrich, St. Louis, MO, USA) solution and each group of rats was administered the corresponding solution intragastrically through gavage once a day at 8:00 a.m. for 28 days. In this study, body weight was measured daily, and food consumption was measured every 7 days. Feces were collected from rats at 26–28 days for the determination of fecal steroids. At the last day, the rats fasted for 18h. Then, the rats were killed under ether anesthesia by drawing blood from the heart. Blood was collected in polypropylene tubes, centrifuged at 3000× *g* for 15 min at 4 °C, and then stored at −80 °C. The liver was collected, weighed and stored at −80 °C for further study.

### 4.3. Analysis of Liver Biochemical Indicators

Initially, the concentrations of 3-hydroxy-3-methyl glutaryl-coenzyme A (HMG-CoA) reductase, cholesterol 7-α hydroxylase (CYP7A1) activity and low-density lipoprotein receptor (LDL-R) in the liver were measured with Rat ELISA Kits (Sen Shellfish Gamma Biotechnology Co., Ltd., Nanjing, China). Total lipids of the liver were extracted by chloroform/methanol solution (2:1, vol/vol), as described by Folch et al. [39]. The liver concentrations of total cholesterol (TC) and triglycerides (TG) were detected by assay kits (Nanjing Jiancheng Biological Engineering Research Institute Co., Ltd., Nanjing, China).

### 4.4. Serum and Fecal Lipid Assays

The serum concentrations of TC, TG and high-density lipoprotein cholesterol (HDL-c) were enzymatically measured with commercial assay kits (Nanjing Jiancheng Biological Engineering Research Institute Co., Ltd., Nanjing, China). TC extracted from fecal matter was measured with commercial assay kits (Nanjing Jiancheng Biological Engineering Research Institute Co., Ltd., Nanjing, China). According to the method reported by Bruusgaard et al. [40] and Malchow-Moller et al. [41], the bile acid was extracted from fecal. Bile acid was measured as follows: 0.5 mL of supernatant was mixed with 3 mL 40% H_2_SO_4_, 1 mL of 0.3% furfural was added after two minutes, and after 30 min of water bath at 65 °C the absorbance at 620 nm was recorded when the pink color developed with maximum intensity.

### 4.5. In Vitro Micelle Experiment

We modified the cholesterol micelle method reported by Kirana et al. slightly [42]. Lipids (0.1 mmol/L cholesterol, 0.1 mmol/L oleic acid, 0.05 mmol/L monooleic acid and 0.06 mmol/L L-α-phosphatidylcholine) were dissolved in chloroform and dried with nitrogen. Then, the lipid mixture was combined with taurocholic acid solution, which was dissolved in 15 mmol/L phosphate buffer/NaCl (pH = 7.4). The micelle suspension was sonicated for 3 min with an ultrasonicator and then incubated in an incubation shaker overnight at 37 °C. Then, 10 mg of casein, WG and WGP were, respectively, added into a 1 mL mixture, ultrasonicated for 3 min and then incubated for 2 h at 37 °C. The solution was separated by a 0.22-micron microporous membrane and 50 µL of filtrate was used for cholesterol determination by a total cholesterol kit (Nanjing Jiancheng Biological Engineering Research Institute Co., Ltd., Nanjing, China). A UV-visible spectrophotometer (Shanghai Zhicheng Analytical Instrument Manufacturing Co., Ltd., Shanghai, China) was used to record the absorbance at 510 nm. The concentrations were calculated based on the standard curve established by the cholesterol calibration standard.
Inhibition ability (%) = [(C0 − C1)/C0] × 100
where C0 represents the cholesterol concentration of the original micelles and C1 is the cholesterol concentration of the micelles with sample fractions.

### 4.6. Antioxidant Activities Assays of Serum and Liver

The total antioxidant capability (T-AOC), superoxide dismutase (SOD), glutathione peroxidase (GSH-PX), catalase (CAT), and malondialdehyde (MDA) of serum and liver were measured with assay kits (Nanjing Jiancheng Biological Engineering Research Institute Co., Ltd., Nanjing, China).

### 4.7. Statistical Analysis

All experimental dates are expressed as the means ± SEMs (standard error of mean). The data of the rats were analyzed by using SPSS software version 21(IBM software, New York, NY, USA). Data were analyzed by one-way analysis of variance (ANOVA), followed by LSD (L) test and *p* < 0.05 was deemed to be a significant difference.

## 5. Conclusions

In summary, we investigated the hypolipidemic effect of WG and WGP from wheat in high-fat diet-induced rats. The treatment of WGP significantly increased the concentration of fecal bile acids and decreased the levels of TC and non-HDL-c in serum (*p* < 0.05). Such benefits were associated with the WGP-mediated downregulation of cholesterol micellar solubility and upregulation of enzymes involved in cholesterol metabolism, including CYP7A1 and LDL-R. Furthermore, WGP significantly decreased the content of MDA and increased the activity of SOD, CAT and the capacity of T-AOC (*p* < 0.05). However, WG did not significantly increase bile acid excretion in feces and decrease the serum levels of TC. These findings indicated that WGP may represent a viable, safe and attractive plant protein approach to attenuate the hyperlipidemia.

## Figures and Tables

**Figure 1 molecules-27-02260-f001:**
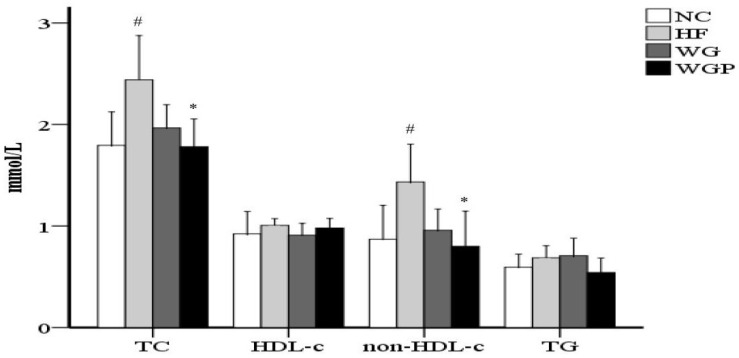
Wheat germ and wheat germ protein intake ameliorated the serum lipid profile in high-fat diet-induced rats. HDL-c, high-density lipoprotein cholesterol; non-HDL-c, non-HDL cholesterol; TG, triglycerides; TC, total cholesterol. Values are presented as means ± SEMs (*n* = 5). ^#^
*p* < 0.05 vs. NC group. * *p* < 0.05 vs. HF group.

**Figure 2 molecules-27-02260-f002:**
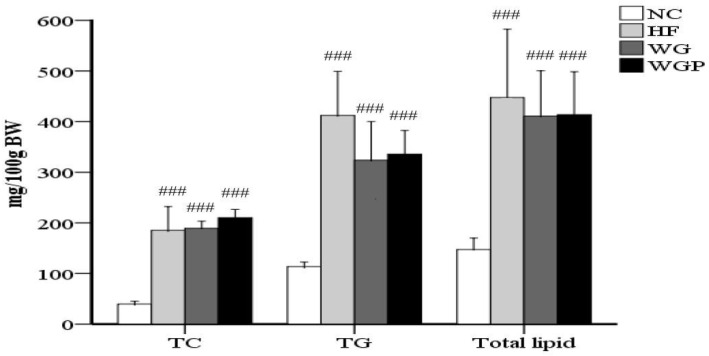
The effects of wheat germ and wheat germ protein on the liver lipid profile of high-fat diet-induced rats. TG, triglycerides; TC, total cholesterol. Values are presented as means ± SEMs (*n* = 5). ^###^
*p* < 0.001 vs. NC group.

**Figure 3 molecules-27-02260-f003:**
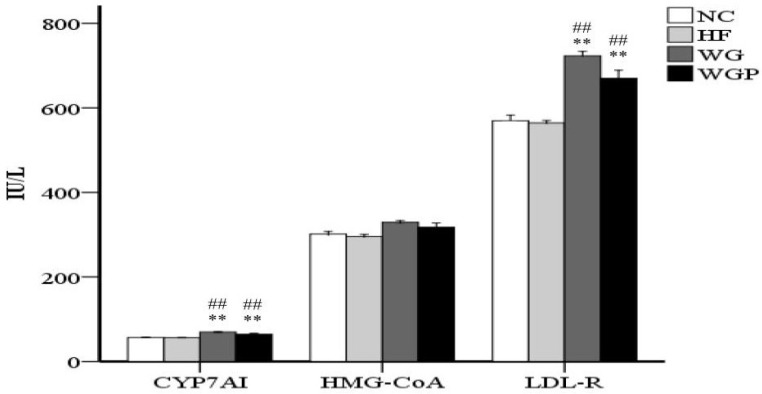
Wheat germ and wheat germ protein modulated the enzyme concentrations of CYP7A1, HMG-CoA and LDL-R in high-fat diet-induced rats. Values are presented as means ± SEMs (*n* = 5). ^##^
*p* < 0.01 vs. NC group. ** *p* < 0.01 vs. HF group.

**Figure 4 molecules-27-02260-f004:**
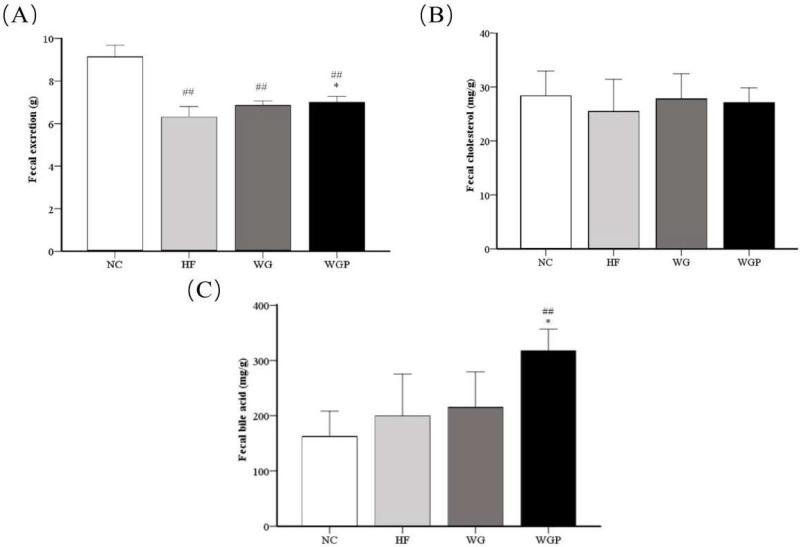
Analysis of fecal excretion (**A**), fecal cholesterol (**B**) and fecal bile acid (**C**) in high-fat diet-induced rats after 4 weeks. Values are presented as means ± SEMs (*n* = 5). ^##^
*p* < 0.01 vs. NC group. * *p* < 0.05 vs. HF group.

**Figure 5 molecules-27-02260-f005:**
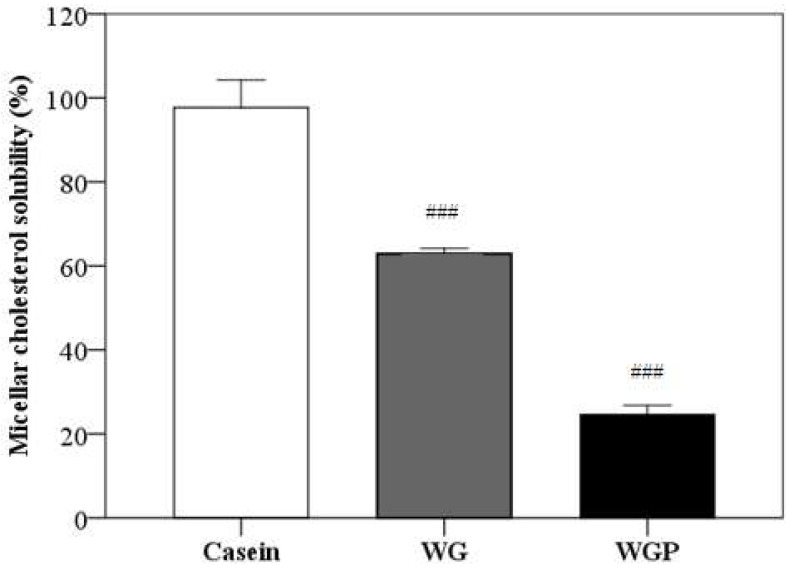
Effects of casein (10 g/L), WG (10 g/L) and WGP (10 g/L) on the micellar solubility of cholesterol in vitro. Values are presented as means ± SEMs (*n* = 4). ^###^
*p* < 0.001 vs. casein.

**Figure 6 molecules-27-02260-f006:**
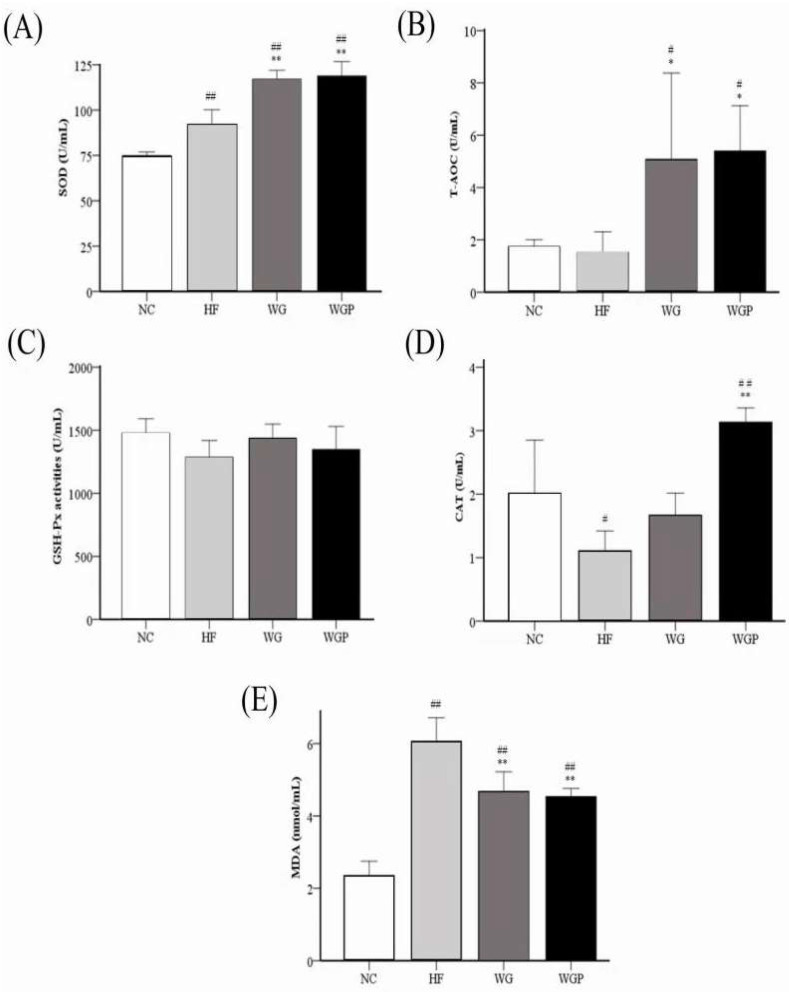
Effect of wheat germ and wheat germ protein on serum antioxidant status, SOD (**A**), T-AOC (**B**), GSH-Px (**C**), CAT (**D**) and MDA (**E**), in the high-fat diet-induced rats. Values are presented as means ± SEMs (*n* = 5). ^#^
*p* < 0.05, ^##^
*p* < 0.01 vs. NC group. * *p* < 0.05, ** *p* < 0.01 vs. HF group.

**Figure 7 molecules-27-02260-f007:**
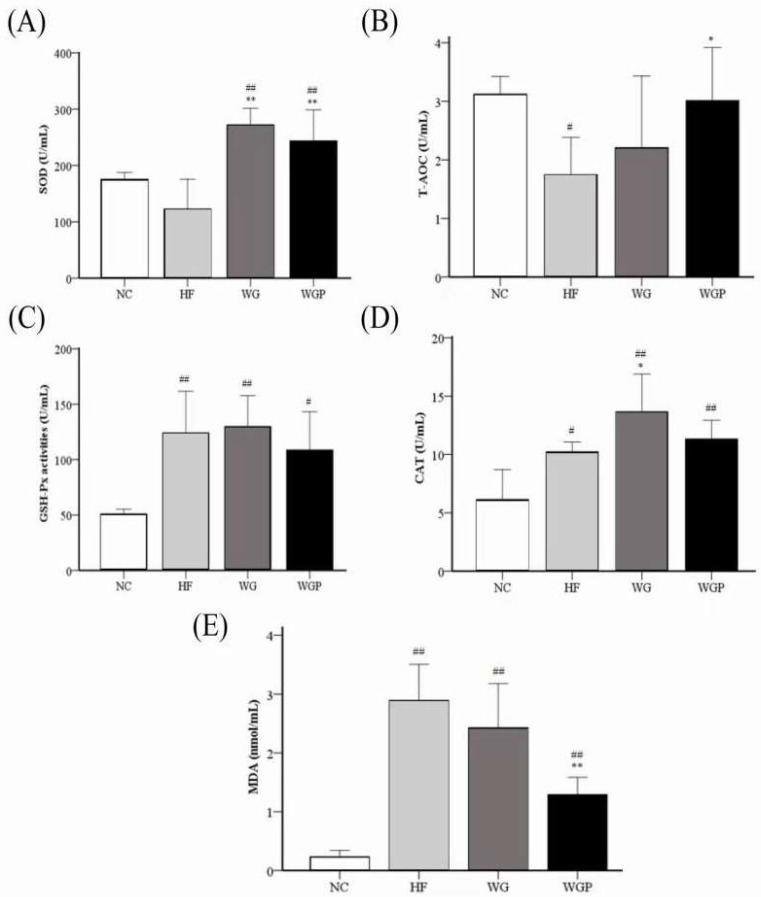
Effect of wheat germ and wheat germ protein on liver antioxidant status, SOD (**A**), T-AOC (**B**), GSH-Px (**C**), CAT (**D**) and MDA (**E**), in the high-fat diet-induced rats. Values are presented as means ± SEMs (*n* = 5). ^#^
*p* < 0.05, ^##^
*p* < 0.01 vs. NC group. * *p* < 0.05, ** *p* < 0.01 vs. HF group.

**Table 1 molecules-27-02260-t001:** Effect of wheat germ and wheat germ protein on body weight, liver weight and food intake in high-fat diet-induced rats. Values are presented as means ± SEMs (*n* = 5). ^#^
*p* < 0.05, ^##^
*p* < 0.01 vs. NC group.

		Experimental Groups		
Parameters	NC	HF	WG	WGP
Initial body weight (g)	158.9 ± 3.5	159.9 ± 2.6	159.3 ± 2.7	157.7 ± 1.8
Final body weight (g)	355.1 ± 8.8	381.8 ± 12.9	363.4 ± 7.7	386.6 ± 11.7 ^#^
Body weight gain (g)	207.1 ± 11.7	221.8 ± 11.5	204.2 ± 6.2	228.9 ± 12.2
Food intake on days 16–17 (g/d)	29.5 ± 5.4	29.0 ± 2.7	30.2 ± 1.4	31.1 ± 1.2
Food intake on days 25–26 (g/d)	31.2 ± 2.8	31.3 ± 2.5	27.6 ± 1.6 ^#^	32.6 ± 1.6
Liver relative weight (%)	2.8 ± 0.1	3.5 ± 0.1 ^##^	3.5 ± 0.1 ^##^	3.4 ± 0.1 ^##^

Liver relative weight (%) = (liver weight/final body weight) × 100%.

**Table 2 molecules-27-02260-t002:** Amino acid compositions of the WG and WGP.

Amino Acid	WG (g/100 g)	WGP (g/100 g)
Asp	3.08	6.76
Thr	1.47	3.23
Ser	1.49	3.85
Glu	4.83	11.30
Pro	1.38	3.41
Gly	2.02	4.75
Ala	2.23	4.90
Val	1.71	4.51
Met	0.54	1.42
lle	1.11	3.02
Leu	2.21	5.93
Tyr	0.96	2.67
Phe	1.28	3.55
His	0.88	2.40
Lys	2.40	5.74
Arg	2.97	7.56
Trp	0.28	0.73
Cys	0.044	0.063

## Data Availability

Not applicable.

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
