# Peer review of "The Hypolipidemic and Antioxidant Activity of Wheat Germ and Wheat Germ Protein in High-Fat Diet-Induced Rats"

_molecules, 2022, doi:10.3390/molecules27072260_

Round 1
Reviewer 1 Report
The manuscript was well written and presented in a neat way. The objective of the present investigation was explained well in the introduction part. Methods adopted for the pharmacological study was found to be appropriate. Relevant literature support is given in the discussion part. I have found some typographical errors and formatting errors in the manuscript. The authors are asked to review the manuscript.
- In results section, a foot note should be included under table no.1 which includes the number of animals/group, presentation of data (Mean ± SEM or SD), method of statistical analysis used and level of significance. If no significance is observed, it should be denoted as NS (Non-significance). The table and figures should be self explanatory.
- In Figure 1, the authors mentioned that different superscript letter indicate statistical significance (P<0.05), but in the bar diagram, different level of significance (a, ab, b) was observed. If the level of significance is same, then what is the meaning of different alphabets in the figure? Moreover the authors should mention the method of comparison used for the analysis (One Way ANOVA followed by?).
- Line no. 95, it is not H group (It is HF) group.
- In Figure 2, no alphabets were given, but in the foot note the level of significance was mentioned. It should be corrected.
- In Figure 3 also the authors did the same mistake and it should be corrected.
- In line no. 117, it is HF group not H group.
- Under Figure 4A, the superscripts are given, but it is not clearly explained.
- Line no. 270, HCl, not HCL
- Under section 4.2, the authors are encouraged to include the Ethics Committee for Animal Study approval number in the manuscript.
- All the chemicals molecular formula should be given in standard format.
Author Response
First of all, Thank you for your time! I appreciate the work you have down!
Please see the attachment.

Reviewer 2 Report
Review: molecules-1558788The hypolipidemic and antioxidant activity of wheat germ and 2 wheat germ protein in high-fat diet-induced rats 3 Cong Liu 1, † , Yi Sun 1, † , Lei Yang 2 , Yuxian Chen 3 , Rigala Ji 1 , Hao Wang 4 , Jinhong Shi 1, *and Jilite Wang 1, *
The authors of the manuscript have significant experience in studying the effect of exogenous salicylic acid on plant cell metabolism, as evidenced by their previous articles on this topic.
The work is well planned and performed at the modern methodological level. The conclusions of the manuscript are fully confirmed by the experimental data obtained by the authors.
The paper uses references that correspond to the topic of the manuscript
The article fully corresponds to the topic of the journal and can be published in molecules
Author Response

(The authors gave the same response as above.)

Reviewer 3 Report
The manuscript entitled “The hypolipidemic and antioxidant activity of wheat germ and 2 wheat germ protein in high-fat diet-induced rats” is a well-organized manuscript reporting the effects of wheat germ protein and wheat germ on cholesterol metabolism, antioxidant activities, and serum and hepatic lipids in rats fed a high-fat diet through gavage
Following clarification/ changes are needed to be done on the manuscript:
1-the paper suffers from the lack of literature comparing wheat germ and the action of wheat fiber regarding its hypolipidemic action
2-The authors need to give a reasonable cause as the idea of this study. Is using only , fifteen rats divided into 3 groups enough for comparison?
Author Response
Reviewer 3
The manuscript entitled “The hypolipidemic and antioxidant activity of wheat germ and 2 wheat germ protein in high-fat diet-induced rats” is a well-organized manuscript reporting the effects of wheat germ protein and wheat germ on cholesterol metabolism, antioxidant activities, and serum and hepatic lipids in rats fed a high-fat diet through gavage
Following clarification/ changes are needed to be done on the manuscript:
1-the paper suffers from the lack of literature comparing wheat germ and the action of wheat fiber regarding its hypolipidemic action
2-The authors need to give a reasonable cause as the idea of this study. Is using only , fifteen rats divided into 3 groups enough for comparison?
Dear reviewer:
First of all, we would like to thank you for giving us the opportunity to revise our manuscript Entitled “The hypolipidemic and antioxidant activity of wheat germ and wheat germ protein in high-fat diet-induced rats”, molecules-1558788.Thank you for your time! I appreciate the work you have down!
Respose 1:
In our studies, WG did not significantly reduce serum TC in rats fed a high-fat diet, which may be related to the duration of the trial, so wheat germ dietary fiber was not discussed separately. It has been reported in the literature that wheat germ has hypolipidemic effect due to its richness in polysaccharides, dietary fiber and polyphenols, but wheat germ protein is ignored. Through this experiment, the hypolipidemic and antioxidant effects of wheat germ protein and their mechanisms were determined for the first time.
Respose 2:
In recent articles, many researchers have exploded the hypolipidemic, antiaging, antibacterial and antioxidant effects of wheat germ. These results indicate that WG as a biological substance should be studied in-depth to examine the various biologically active ingredients contained in it. The above content has been explained in lines 38-42 of Introduction, you can check this in lines 38-42.
The selection of 5 animals in this experiment is based on the findings of previous research, and is also based on the consideration of animal welfare of the smallest sample. Generally, the minimum sample size for researchers doing biological research is 5.
Reviewer 4 Report
First af all, I appreciate your submitting of your manuscript to this journal « Molecules » and for giving us the opportunity to consider your work.
After revewing the manuscript, I suggest the following corrections should be done regarding the structure, the content and form of the text.
Abstract : You have to arrange your abstract on Background, objective, methodology, results and conclusion
Introduction : well structured
Results :
As indicated in table 1, you notified any significant differences, regarding growth parameters (body weight gaing), However, you in the results related to the biochemical parameters, you found significant differences, How can you nexplain these paradoxical results. Tese observation should inserted in discussion section.
What do you mean by the letters a,b and ab. Did you mean the significace, in this you have to mention this in your legend. Check this in all figures of your manuscript
Methodology : need major revision as indicated in following comments
Line 284 : In animal experiment section, you talk about 3 groups (HF,WG,WGP). The designation of the groups is incomprehensible, normally the WG and WGP groups must be feeded by HF,by this we can have an idea on the products tested (WG and WGP)
Your methodology requires some major changes, if you want to know the effect of the treatment of animals, you have to add, at least, two groups: normal control groups which has not been feed either by HF or by the product to be tested (Essential). Additionally, you have to add another positive group treated by medicament, as standard.
How you got the amino acid values shown in table 2. By which method or from which reference you were able to obtain these values.
On which references did you rely for determinoing the doses, (300 mg/kg) for both WG and WGP, choosen for your experiment ??????
We are very worried about the body weight of the rats used in your experiment (between 70g and 80g). It seems very low weight. You do not think that this low weight could distort your results, or at least could have an effect on your results
You should justify the use of male rats instead of female.
Basically, female groups must be included in your experiments, by this mean you can have a general idea of the effect of these products on a given animal species and not the male genus alone

Author Response
Dear reviewer:
First of all, we would like to thank you for giving us the opportunity to revise our manuscript Entitled “The hypolipidemic and antioxidant activity of wheat germ and wheat germ protein in high-fat diet-induced rats”,molecules-1558788.Thank you for your time! I appreciate the work you have down!
Abstract :
- You have to arrange your abstract on Background, objective, methodology, results and conclusion
Response 1:
Thanks, for your kindly remind! I arranged my abstract on objective, methodology, results and conclusion. This time, I added background to the abstract.
Results :
1.As indicated in table 1, you notified any significant differences, regarding growth parameters (body weight gaing), However, you in the results related to the biochemical parameters, you found significant differences, How can you explain these paradoxical results. Tese observation should inserted in discussion section.
2.What do you mean by the letters a,b and ab. Did you mean the significace, in this you have to mention this in your legend. Check this in all figures of your manuscript
Response 1:
Thanks, for your kindly remind! In the authors' published papers, there was no significant difference in body weight between the high-fat group and the food-derived protein or peptide group during the 2 or 3 weeks of the experiment (References 1 and 2). This may be related to the experimental period and animal feed composition, on the other hand, it may indicate that food-derived proteins or peptides have no effect on the health of experimental rats.
- Jilite Wang, Masaya Shimada, Yukina Kato, Mio Kusada & Satoshi Nagaoka. Cholesterol-lowering effect of rice bran protein containing bile acid-binding proteins,Bioscience, Biotechnology, and Biochemistry,2015,79:3,456-461,DOI: 10.1080/09168451.2014.978260
- Arata Banno, Jilite Wang, Kenji okada, Ryosuke Mori, Maihemuti Mijiti&Satoshi nagaoka.Scientific Reports,(2019) 9:19416,https://doi.org/10.1038/s41598-019-56031-8
In most studies, high cholesterol diets have little effect on body weight, more on blood lipids and liver lipid levels.
(1) DOI:10.1089/jmf.2020.0141. In this article, the author explored hypolipidemic effects and cholesterol metabolism modulation of Jaboticaba (Myrciaria cauliflora) peel in diet-Induced NAFLD rat model. In this article, there was no significant difference in the body weight of the mice and the blood lipid levels of the mice were significantly reduced.
(2)https://doi.org/10.1016/j.fbio.2014.08.001. In this article, the author used adult rats to explore if rice protein can regulate HDL metabolism-related gene expression and enzyme activity. In this article, there was no significant difference in the body weight of the mice and the blood lipid levels of the mice were significantly reduced.
(3)Doi:10.1088/1755-1315/512/1/012097. In this article, the author investigated the hypoglycemic and hypolipidemic effects and the underlying mechanism of ethanol extracted from the aerial part (AUC)and underground part of Urtica cannabinaL(UUC) using alloxan-induced hyperglycemic mice model. In this article, there was no significant difference in the body weight of the mice and the blood lipid levels of the mice were significantly reduced.
Response 2:
Thanks, for your kindly remind! I have corrected the superscript letters.
Methodology :
1.Line 284 : In animal experiment section, you talk about 3 groups (HF,WG,WGP). The designation of the groups is incomprehensible, normally the WG and WGP groups must be feeded by HF,by this we can have an idea on the products tested (WG and WGP)
Response 1:
Thanks, for your kindly remind! Feeds for the WG and WGP groups may not have been well described in the original paper. The WG and WGP groups were fed with high-fat diet in this study. This information has been supplemented in the paper, see line 286 for details.
2.Your methodology requires some major changes, if you want to know the effect of the treatment of animals, you have to add, at least, two groups: normal control groups which has not been feed either by HF or by the product to be tested (Essential). Additionally, you have to add another positive group treated by medicament, as standard.
Response 2:
Thanks, for your kindly remind! It has been reported that wheat germ has hypolipidemic and antioxidant effects. Feeding different levels of wheat germ caused significantly decreased in serum levels of TL, TG, TC, LDL-C, VLDL-C in Rats Fed Cholesterol-Containing Diet. However, the mechanism of action remains unclear (References 1 and 2). In this experiment, we mainly elucidate the mechanism of wheat germ and wheat germ protein regulating blood lipids and oxidative stress. Therefore, when designing the experiment, more emphasis was placed on the mechanism of lowering blood lipids and antioxidative effects, rather than confirming the results reported by others. We’re appreciate that you remind us this important question. Thanks for your time!
- Rezq, A.; Mahmoud, M. Preventive Effect of Wheat Germ on Hypercholesteremic and Atherosclerosis in Rats Fed Cholesterol-Containing Diet. Pakistan Journal of Nutrition 2011, 10, doi:10.3923/pjn.2011.424.432.
- Liaqat, H.; Kim, K.J.; Park, S.Y.; Jung, S.K.; Park, S.H.; Lim, S.; Kim, J.Y. Antioxidant Effect of Wheat Germ Extracts and Their Antilipidemic Effect in Palmitic Acid-Induced Steatosis in HepG2 and 3T3-L1 Cells. Foods 2021, 10, doi:10.3390/foods10051061.
3.How you got the amino acid values shown in table 2. By which method or from which reference you were able to obtain these values.
Response 3: Thanks, for your kindly remind! The amino acid composition is tested by L-8900 Amino Acid Analyzer (Hitachi, Japan). I added this information in my manuscript. See line 285 for details.
4.On which references did you rely for determinoing the doses, (300 mg/kg) for both WG and WGP, choosen for your experiment ??????
Response4:
Thanks, for your kindly remind! In many papers, researchers choose doses of 200mg/kg, 400mg/kg or 800mg/kg to study the hypolipidemic effect of bio-substances in rats. I listed some literatures for you to check.
- https://doi.org/10.1016/j.fct.2019.110663; In this article, the author chose the dose of 200mg/kg and 400mg/kg body weight to investigate the hypolipidemic effect of polysaccharides from Fortunellamargarita (Lour.) Swingle (FMPS) in hyperlipidemic rats.
- https://doi.org/10.1016/j.ijbiomac.2018.12.082; In this article, the author chose the dose of 200mg/kg and 400mg/kg body weight to investigate hypolipidemic and antioxidant effects of Pine needle polysaccharide (PNP) from Pinus massoniana in high-fat diet (HFD)-induced mice.
- https://doi.org/10.1016/j.biopha.2021.111219. In this article, the author chose the dose of 150 mg/kg, 380 mg/kg and 540 mg/kg body weight to reveal hypoglycemic and hypolipidemic mechanism of Xiaokeyinshui extract combination on streptozotocin-induced diabetic mice. This experiment refers to the above papers, we selected a dose of 300 mg/kg body weight to explore the hypolipidemic and antioxidant activity of wheat germ and wheat germ protein in high-fat diet-induced rats.
5.We are very worried about the body weight of the rats used in your experiment (between 70g and 80g). It seems very low weight. You do not think that this low weight could distort your results, or at least could have an effect on your results
Response 5:
Thanks, for your kindly remind! In the article published by author Jilite Wang, Scientific reports.2019,9:19416 /https://doi.org/10.1038/s41598-019-56031-8, researchers used 4-week-old rats with a body weight of 70 g to Identify a novel cholesterol-lowering dipeptide, phenylalanine-proline (FP), and explore its down-regulation of intestinal ABCA1 in hypercholesterolemic rats. This experiment refers to the above papers, and selects male rats with a body weight of 70g. After acclimation to a commercial nonpurified diet for 7 days, body weight was around 159g (see table 1 for details). During the breeding process, it was found that there was no abnormality in the growth performance. We feed them for 4 weeks and at last the body weight was around 380g. And the indicators’ differences between individuals of the same group are small, so we are sure that low body weight has no effect on the experimental results.
6.You should justify the use of male rats instead of female.
Basically, female groups must be included in your experiments, by this mean you can have a general idea of the effect of these products on a given animal species and not the male genus alone
Response 6:
Thanks, for your kindly remind! According to our investigation, most of the gender-based considerations and analyses are often ignored for preclinical research, but for drug efficacy evaluation experiments, animals of both genders are often selected due to differences in sensitivity between male and female animals. Changes in estrogen often affect the metabolism of animals, including lipid metabolism and oxidative stress defense (refs 1, 2, 3 below). In this paper, we discuss The hypolipidemic and antioxidant activity of wheat germ and wheat germ protein in high-fat diet-induced rats for the first time. Considering the experimental cost and particularity, we preferred male rats. When functional food development and its clinical research are conducted in the future, as you said, the particularity of the population must be considered and the female population must be included.
1)Allen J. Schreiber,Francis R. Simon. Estrogen-Induced Cholestasis: Clues to Pathogenesis and Treatment.HEPATOLOGY.1983.3(4),607-613.https://doi/org/10.1002/hep.1840030422
2)Claes-Henrik Floren,Rampratap S.Kushwaha,William R.Hazzard,John J.Albers. Estrogen-induced increase in uptake of cholesterol-rich very low density lipoproteins in perfused rabbit liver.Metabolism.1981.30(4):367-375. https://doi.org/10.1016/0026-0495(81)90117-7
3)K. Weber and R. G. Erben.Differences in triglyceride and cholesterol metabolism and resistance to obesity in male and female vitamin D receptor knockout mice.Journal of animal physiology and animal nutrition.2013.97:675-683.DOI: 10.1111/j.1439-0396.2012.01308.x

Round 2
Reviewer 4 Report
Unfortunately, the authors misunderstood the majority of the comments. He didn’t reply to any comment,See the details Below :
Abstract :
- You have to arrange your abstract on Background, objective, methodology, results and conclusion
Response 1:
Thanks, for your kindly remind! I arranged my abstract on objective, methodology, results and conclusion. This time, I added background to the abstract.
Any changing in the abstract, see the manuscripts
Introduction
The botanical name of the plant (Triticum aestivum L.) should be written in Italic form like (Triticum aestivum L. You have to checke this name over all the manuscript.
Results :
1.As indicated in table 1, you notified any significant differences, regarding growth parameters (body weight gaing), However, you in the results related to the biochemical parameters, you found significant differences, How can you explain these paradoxical results. These observations should be inserted in discussion section.
Response 1:
Thanks, for your kindly remind! In the authors' published papers, there was no significant difference in body weight between the high-fat group and the food-derived protein or peptide group during the 2 or 3 weeks of the experiment (References 1 and 2). This may be related to the experimental period and animal feed composition, on the other hand, it may indicate that food-derived proteins or peptides have no effect on the health of experimental rats.
Your revision is not convincing, but it puts a lot of doubt on the reliability of your results. Indeed, you mention in the text dLine xxxxxx that there are no significant differences, in the legend of your table 1, you indicated, with a letter (a) the significance everywhere. However in the previous version of your manuscripts, you didn’t mentione any significance, in table 1 ???????, it is very surprising ?????
2.What do you mean by the letters a,b and ab. Did you mean the significace, in this you have to mention this in your legend. Check this in all figures of your manuscript.
The letters mentioned in the figures have no meaning for the readers. For example the letter (a) signifies the meaning between which or which group. This must be mentioned in the legend for all the letters. your results are very confusing, You have to explaine the significance obtained, between which groups. As presented in your manuscripts, the reader is really lost. You have to give more explanation in your legend.
Methodology
2.Your methodology requires some major changes, if you want to know the effect of the treatment of animals, you have to add, at least, two groups: normal control groups which has not been feed either by HF or by the product to be tested (Essential). Additionally, you have to add another positive group treated by medicament, as standard.
Response 2:
Thanks, for your kindly remind! It has been reported that wheat germ has hypolipidemic and antioxidant effects. Feeding different levels of wheat germ caused significantly decreased in serum levels of TL, TG, TC, LDL-C, VLDL-C in Rats Fed Cholesterol-Containing Diet. However, the mechanism of action remains unclear (References 1 and 2). In this experiment, we mainly elucidate the mechanism of wheat germ and wheat germ protein regulating blood lipids and oxidative stress. Therefore, when designing the experiment, more emphasis was placed on the mechanism of lowering blood lipids and antioxidative effects, rather than confirming the results reported by others. We’re appreciate that you remind us this important question. Thanks for your time!
Your answer is not convincing, my objection in this comment is to add another group “normal control ”, I mean a group of healthy rat. Without this group the comparison would remain irrelevant.
